# Recruiting on the Spot: A Biodegradable Formulation for Lacewings to Trigger Biological Control of Aphids

**DOI:** 10.3390/insects10010006

**Published:** 2019-01-05

**Authors:** Joakim Pålsson, Gunda Thöming, Rodrigo Silva, Mario Porcel, Teun Dekker, Marco Tasin

**Affiliations:** 1Department of Plant Protection Biology, Swedish University of Agricultural Science, 230 53 Alnarp, Sweden; Joakim.palsson@slu.se (J.P.); mporcel@agrosavia.co (M.P.); teun.dekker@slu.se (T.D.); 2NIBIO, Norwegian Institute of Bioeconomy Research, Postbox 115, NO-1431 Ås, Norway; Gunda.Thoeming@nibio.no; 3Isca Technologies Inc., 1230 Spring St., Riverside, CA 92507, USA; rodrigo.silva@iscatech.com

**Keywords:** apple, barley, *Chrysoperla carnea*, herbivory induced plant volatiles, integrated pest management, semiochemicals, ecological intensification

## Abstract

Upon herbivory, plants release herbivore-induced plant volatiles (HIPVs), which induce chemical defenses in the plant as well as recruit natural enemies. However, whether synthetic HIPVs can be employed to enhance biological control in a cultivated crop in the field is yet to be explored. Here we show that a biodegradable formulation loaded with induced and food-signaling volatiles can selectively recruit the common green lacewing, *Chrysoperla carnea*, and reduce pest population under field conditions. In apple orchards, the new formulation attracted lacewing adults over a 4-week period, which correlated well with independent assessments of the longevity of the slow-release matrix measured through chemical analyses. In barley, lacewing eggs and larvae were significantly more abundant in treated plots, whereas a significant reduction of two aphid species was measured (98.9% and 93.6% of population reduction, for *Sitobion avenae* and *Rhopalosiphum padi*, respectively). Results show the potential for semiochemical-based targeted recruitment of lacewings to enhance biological control of aphids in a field setting. Further research should enhance selective recruitment by rewarding attracted natural enemies and by optimizing the application technique.

## 1. Introduction

Intensification of food production has been at odds with ecological sustainability. Among the unintended side effects are the loss of biodiversity, the simplification of the landscape, and an increased reliance on pesticides [1]. While these production systems favor high yields, corresponding repercussions on multitrophic relationships, such as pest’s natural enemy interactions, have been significant [2]. Ecological intensification is proposed as an approach to restore multifunctionality in agro-ecosystems, while supporting high productivity levels [3].

Enhancing biological control of insect pests through ecological processes is among the objectives of the “ecological paradigm shift” [3]. While an increasing body of evidence demonstrates how botanical diversity can effectively prevent pest attacks in crops by re-establishing trophic cascades [4], practical knowledge and cultural and economic gaps often impede the adoption of this concept in both high and low-input agricultural systems [5,6]. New methods should thus be developed to gradually transit from current input intensive monoculture to ecologically intensified crops.

Plants developed chemical and physical defenses to counter herbivory, some of which are only expressed upon attack. These inducible defense mechanisms can be particularly advantageous when herbivores are diverse and variable in time and space, and allow plants to optimize resource allocation and tailor the response to the current threats [7]. Among inducible defenses, herbivory-induced plant volatiles (HIPVs) rapidly prime surrounding tissue and neighboring plants and recruit natural enemies from a distance. In order to induce systemic defense in the whole plant, HIPVs are released from the damaged area into the atmosphere [8,9]. A faster induction at the individual plant level may be achieved through the atmosphere, especially in large plants such as trees. HIPVs additionally allow the releasing plant to communicate with other trophic levels, attracting predators and parasitoids [10]. As the emission of HIPVs occurs exclusively upon herbivory, they provide an honest cue to both specific and generalist third trophic level arthropods. Given such properties, synthetic HIPVs may be used as a tool for pest management in agriculture [11].

Several suggestions on how to utilize HIPVs in crops are reported in literature, either as a way of monitoring beneficial arthropods within a crop or to attract natural enemies in an attempt to directly control pest populations [12]. Indeed, HIPVs have been found to attract natural enemies belonging to different families, including hoverflies (Diptera: Syrphidae), predatory bugs (Heteroptera), ladybirds (Coleoptera: Coccinellidae), predatory mites (Mesostigmata), parasitic wasps (Hymenoptera), and green lacewings (Neuroptera: Chrysopidae) [11]. Because the composition of the volatile blend was found to fine-tune the range of attracted natural enemies, information on possible synergies among volatiles with a diverse ecological function could be used to specifically modulate behaviors of such beneficial arthropods [13]. For example, acetic acid (AA), which is often considered a sugar signaling compound, enhanced attraction of a number of insect species to plant volatiles [14,15]. Similarly, phenylacetaldehyde (PAA), which is a nectar-signaling floral as well as an induced volatile [16,17], was reported to increase predator attraction when added to other plant volatiles [16]. The synergy between food-signalling cues and classical inducible compounds, such as methyl salicylate (MS), which is attractive for several predatory groups [18], may be particularly interesting in species where biological control is not directly exerted by the attracted adults, but rather by their offspring. One such biological control agent is the common green lacewing, *Chrysoperla carnea s.l.*, adults of which feed on nectar and pollen, while the larvae are generalist predators [19]. Several compounds attracted significant amounts of lacewings (i.e., Acetophenone, 2-phenylethanol (PE), PAA) in cherry, apricot, apple, pear, and walnut orchards [16,20,21]. Interestingly, a blend of MS, PAA, and AA not only attracted adults, but was also capable of increasing the oviposition rate in the vicinity of the releasing point, possibly thanks to the presence of food signalling volatiles [22]. In addition, this blend significantly increased lacewing density in overwintering boxes [22]. Whether this blend is capable of enhancing pest control in the surrounding vegetation remains to be investigated.

We hypothesized that a novel formulation consisting of plant and food odors embedded in a biodegradable matrix can attract lacewing from surrounding vegetation and induce oviposition on the plant. We further hypothesized that this recruitment of adults would contribute to the suppression of aphid populations.

Common green lacewing attraction from a distance and release of volatiles from emitting devices were evaluated in apple orchards (*Malus domestica* Borkhausen). However, the assessment of biological control was carried out in a cereal field, due to a higher and much more homogeneous pest attack in comparison to orchards. Parameters such as larval lacewing density and population level of other natural enemies were measured in the system of barley (*Hordeum vulgare* L.), two hemipteran herbivores, *Sitobion avenae* (F.) and *Rhopalosiphum padi* (L.), and the common green lacewing as a generalist predator. In order to propose a method with a low environmental impact, we compared the efficacy of a standard polyethylene-based bag odor dispenser with a newly assembled prototype based on a biodegradable matrix.

## 2. Materials and Methods

### 2.1. Volatile Releasing Formulations

The reference device was purchased from Csalomon (Plant Protection Institute, MTA ATK, Budapest, Hungary). It consisted of a cotton wick loaded with a 3-component blend of MS, PAA, and AA in a 1:1:1 ratio with a total load of 300 mg/device. The wick was placed into a sealed polyethylene bag, through which volatiles were slowly released (hereafter referred to as PE bag). In order to facilitate its use in the field, this device was delivered with a pre-attached plastic strip to be stapled into the canopy, or to a holding stick. We selected this formulation as a benchmark because field data on lacewing attraction and oviposition using this formulation were available at the time of our study [20]. The new formulation was a novel product prepared in co-operation with ISCA Technologies (ISCA Technologies Inc., Riverside, CA, USA) and Bio-Innovate AB (Lund, Sweden). It consisted of a biologically inert, biodegradable wax-water emulsion releasing paste loaded with the above-described blend at a concentration of 300 mg/mL. A single release point for this product constituted a 1 mL droplet applied with a plastic syringe (hereafter referred to as paste). Data on the slow-release properties of the paste using other volatile cues are also available [23].

### 2.2. Measurement of Volatile Release

To compare over time the volatile release rate from the two formulations, PE bag and paste were hung within the canopy of apple trees in Alnarp (Lomma, Sweden) at a height of approximately 1.7 m from the ground in the beginning of May 2016. The releasing devices were retrieved from the tree at 1, 7, 14, 21, and 28 days after field exposure (5 devices per date). After collection, they were placed into a 43 mL plastic vial and covered with Parafilm^®^ for headspace collection. After 60 min of stabilization, a solid-phase microextraction (SPME) fiber (Divinylbenzene/Carboxen/Polydimethylsiloxane, Sigma-Aldrich, St Louis, MO, USA) was inserted through the Parafilm for volatile collection. After 15 min, the SPME fiber was retracted and directly inserted into a gas-chromatograph (GC) injector connected to a mass spectrometer (GC-MS 5977A MSD, Agilent technologies, Santa Clara, CA, USA) equipped with a polyethylene glycol column (DB-Wax, length 60 m, diameter 0.25 mm, df 0.25 µm, Agilent Technologies, Santa Clara, CA, USA) for 40 sec. The GC-MS started at 100 °C for 2 min and increased 15 °C each 2 min until 200 °C and then finished at 250 °C for 2 min. Prior to each volatile collection, the SPME device was conditioned in a GC port at 250 °C for 10 min. After headspace collection, each lure was wrapped in aluminum foil and stored at −80 °C until use in the subsequent trapping experiment (see below). Volatiles were identified by comparing their spectra with those published in the reference library NIST05. In addition, AA, MS, and PAA were identified by comparing their retention index with those of synthetic standards.

To estimate the release rate of the main components, microcapillary tubes were partially filled with one of the compounds and placed in the same 43 mL vials described previously [24]. Sampling followed the same protocol as described above. The microcapillary tubes were weighted before and after headspace collection. The weight loss of the microcapillary tubes was then related to the area of the corresponding GC-MS peak to calibrate the amounts released. Of the three main compounds, only AA evaporated in measurable amounts using this protocol, and was therefore used for calibration.

### 2.3. Attraction Longevity

To determine the device’s attraction longevity in the field, aged lures previously collected from the orchard and analyzed were removed from the freezer and placed in McPhail traps (Sanidad Agrícola Econex S.L., Murcia, Spain). Traps were subsequently hung in five apple orchards from the 5th to the 19th of August 2016. Orchards were located in Kivik (Skåne county, Sweden. 55°41′ N, 14°13′ E). Three orchards were organically certified and two were under integrated protection (IP). The orchards (minimum 5 ha surface) were situated a minimum of 1 km apart. In each orchard, 12 McPhail traps were placed in two different circles (diameter 14 m). Each circle was comprised of six traps loaded with either PE bag or paste aged at 1, 7, 14, 21, to 28 days and with a blank trap. The two circles were 30 m apart and at least 10 m from the orchard border. Traps were hung at a height of 1.6 m and 7 m apart. Trap position was randomized within each circle at the start of the experiment. Each trap was inspected twice a week over a two-week period. In order to avoid positional effects, traps were rotated two steps in a clockwise direction within each circle at each inspection (Appendix A). The collected specimens were stored in ethanol (70 vol. %) for species identification and sex determination.

### 2.4. Measurement of Biological Control

A field experiment was conducted in spring barley fields (*H. vulgare*, cv. Helium) in Ås, Norway (59°67′ N, 10°77′ E) in June and July 2016. Either a PE bag or a paste formulation was installed at the center of 25 m^2^ plots (N = 12). Distances between plots were at least 5 m. Three different types of lure applications were tested: (1) PE bag dispenser (300 mg total load) at vegetation height; (2) 1 mL paste-droplet with 300 mg total load on the plant (paste_1x); and (3) 3 mL paste-droplet with 900 mg total load on the plant (paste_3x). Four plots for each of the three formulation types were arranged randomly within the crop. The PE bag dispensers were hung on wooden sticks at approximately vegetation height. The wire enabled weekly adjustment to the height of the dispenser to mirror the vegetation height of the growing barley plants. The paste droplets (1 or 3 mL/plant) were applied with a 100 mL-syringe on leaves in the upper third of the barley plants. At a distance of 400 m, four control plots (25 m^2^) without treatment were installed with at least 5 m between the plots, as mentioned above. The 400 m distance was used to reduce the influence from either of the treatments on the control plots, as range of effect is unknown for these formulations. In the middle of each plot, a marker point (wooden stick, 60 cm) was installed. In each plot, visual inspection of lacewings (eggs and larvae of *C. carnea s.l.*), aphids (nymphs and adults of *S. avenae* and *R. padi*), and other natural enemies (Coccinellidae larvae, Syrphidae larvae, and parasitized aphid mummies) was performed in five differentiated sectors. Sectors were established as dispenser or marker points (=Centre; C), and 30 cm distances in the directions north (N), south (S), west (W), and east (E) of the dispenser or marker points. The observations were performed on the three plants nearest to the five marked points (C, N, S, W, E). Counted lacewing eggs were marked with a small dot on the leaf to avoid repeated counts. At the phenological stage 13 (leaf development, 3 leaves unfolded, 1st of June) of barley plants, the sectors were checked for aphids, lacewings, and other natural enemies (first record). Then the different lure types and marker points were placed in the experimental barley fields. Over an experimental period of eight weeks, the sectors were checked weekly for lacewings, aphids, and other natural enemies, as described above. Dispensers and droplets were replaced once after four weeks. Our observations were carried out within a landscape with wild inter-field vegetation, where lacewings may have had access to floral resources, overwintering sites, and alternative prey.

### 2.5. Data Analysis

Statistical analyses were conducted with RStudio v. 1.0.143 and R software v. 3.3.3 [25], with packages lme4 v. 1.1-12 [26] and LSmeans v. 2.27-61 [27]. Fixed factors were checked for significance with the Wald-test from the car package [28].

The analysis of the release rates was carried out with linear regression models (LRM), including *formulation* (PE bag, Paste) and *age* (1, 7, 14, 21, 28 days) and their interaction as fixed factors. LRM residuals were examined visually with a QQ-plot to ascertain the normality assumption. The model for AA did not present normally distributed residuals and homogeneity of variance and was analyzed with a generalized least squares model (GLS). The amount of AA released was square root transformed and contrast was set to sum-to-zero. Tukey’s test was used for post-hoc testing of the factors *age* and *formulation*.

Cumulated capture of lacewings in the McPhail traps were analyzed with generalized linear mixed models (GLMMs) with a negative binomial distribution. Initially, several exploratory models were built. The models with best fit were used to test for the possible effect of trap position within the plot with each of the formulations analyzed separately as response variables. Both models included *position* as a fixed factor and *orchard* as a random effect. Additionally, to test for the possible attractiveness of unbaited McPhail traps (blanks), GLMMs were used again for both formulations separately, using catches as response variables, *age* or *blank* (1, 7, 14, 21, 28, and blank) as a fixed factor, and *orchard* as a random effect. Dunnett’s test was then used for post-hoc testing of the different ages against the blank. Finally, a GLMM was fitted to test the effect of lures on lacewing catches. The model included catches with both formulations as response variables, *formulation*, *age*, and their interaction as fixed factors, and *orchard* as a random effect. A Tukey’s test was used for pairwise multiple comparisons. In addition, two models (with each formulation analyzed separately) were fitted to test for a possible divergence from a 1:1 sex ratio in trap catches. These GLMMs included a binomial error distribution with a logit-link, *age* as fixed factor, and *orchard* as random effect. In addition to lacewings, non-parasitic wasps (Hymenoptera: Vespidae) were caught in the McPhail traps. The Fisher’s exact test was used with binomial transformed data to determine the effect of formulation (PE bag or Paste versus Blank) and the difference between them (PE bag versus Paste).

The data collected in cereal included the number of lacewing eggs and larvae, aphids, and other natural enemies, and were analyzed with GLMMs with a Poisson or a negative binomial distribution and a log-link. The count of each of these groups was used as the response variables with *treatment* (formulation: PE bag, Paste_x1, Paste_x3, Control), *sector*, interaction between *treatment* and *sector*, and *date* as fixed factors. Additionally, *sector* nested within *plot* was included as a normally distributed random effect. The choice of the most fitting distribution was based on a test for overdispersion. After establishing the significance of the fixed factors, Tukey’s tests were carried out for pairwise comparisons between levels of each factor when necessary. All GLMMs were followed by Wald tests for statistical inference and were validated by inspecting visually the studentized residuals against the fitted values.

## 3. Results

### 3.1. Measurement of Volatile Release

The emission of the loaded compounds decreased over time in both formulations (Figure 1A, Table 1 and Appendix A). Release rates of AA, MS, and PAA were higher from the reference dispenser over the four-week period, except for PAA at day 1 (Figure 1A). Although MS emission was high over the entire period of field exposure and PAA showed an intermediate release rate, AA emission quickly decreased. Estimated release rates calibrated using AA were in the range of 0 to 1.5 mg/day. Beside the three main components, both formulations emitted lower quantities of 2-phenylethanol, benzaldehyde, benzyl acetate, and benzyl alcohol, whereas 2-heptenal and hexanoic acid were released exclusively by the paste (Figure 1B,C). Climatic data were recorded every 15 min by the Lantmet weather station at Alnarp. The temperature at 1.5 m height from the soil was 18.5 ± 0.1 and 9.9 ± 0.1 °C, during day and night, respectively. The wind speed was 4.0 ± 0.0 and 2.4 ± 0.0 m/s, during day and night, respectively. The rainfall during the exposure time of the formulations in the field was 17.4 mm.

### 3.2. Attraction Longevity

A total of 348 adult *C. carnea s.l.* were caught in traps in the attraction experiment. There was no effect of trap position on catch (Table 1). Blank traps caught 4 lacewings. All treatments were significantly different against the blank, except for the 28-day-old paste. Although the reference device attracted in total a higher number of lacewings than the new paste (Table 1), the decline over time in attraction was comparable for most of the dates (Figure 2). Both formulations were equally attractive during the two first weeks and caught the highest number of catches in the newest lure. After 21 days only the reference dispenser remained attractive (Figure 2). Males and females were equally attracted by both dispenser types independently of age, except 1-day-old paste, which attracted significantly more females (Figure 2, Table 1). In addition to lacewings, a total of 89 wasps (Hymenoptera: Vespidae) were caught as by-catches. Although both formulations caught a higher number of wasps than blank traps, no difference between formulations was detected (Table 1).

### 3.3. Measurement of Biological Control

A total of 983 lacewing eggs and 1965 larvae were recorded in the barley fields. The number of eggs and larvae changed over time and differed between treatments (Figure 3A, Table 1). The abundance of immature lacewing stages varied between sectors (Figure 4A, Table 1). The most lacewing eggs and larvae were recorded with the reference formulation, followed by paste_1x and paste_3x. Very low numbers were found in control plots (Figure 4A). Eggs and larvae clustered close to the lure (sector C), while in the control plot they appeared evenly distributed in all sectors. However, this difference is only statistically supported for eggs in treatments with the PE bag formulation and paste_1x, except for the north sector of paste_1x, and for larvae in the same treatments (Figure 4A).

Other natural enemies were also observed in the field, where we encountered a total of 95 adult ladybirds (Coccinellidae: *Coccinella septempunctata* L.), 22 hoverfly larvae (Syrphidae), and 72 parasitized aphid mummies. These natural enemies occurred in lower numbers than lacewing eggs and larvae in all treatments (Figure 4A,B). Ladybird adults and hoverfly larvae were recorded in similar numbers in all treatments and in the control, whereas aphid mummies were more abundant in the control than in the rest of the treatments (Figure 4B).

Of the two different species of aphids (*S. avenae* and *R. padi*), *S. avenae* was the most abundant (Figure 3C and Figure 4C). The abundance of both aphid species changed over time and differed between treatments. Both aphid species were more abundant in the control as compared to any of the treatment plots, irrespective of sector (Figure 4C). *S. avenae* was less abundant in the paste_1x than paste_3x, whereas *R. padi* abundance was independent of dollop density.

Lacewings were the first natural enemies recorded in the season (Figure 3A). Their eggs were observed in the second week of the field trial, followed by larvae in the third week. Other natural enemies, primarily ladybird larvae, were observed in week six (Figure 3B). Aphids were first detected in the second week in similar amounts in all treatments (Figure 3C).

## 4. Discussion

Adult attraction of *C. carnea* from a distance as well as reduction of aphid population were triggered by a blend of an induced compound and two food-signalling kairomonal cues in our field experiments. Previous studies showed that the blend used here attracted lacewings, particularly of the *Chrysoperla* species complex [29], and stimulated their oviposition around the formulation in cherry, apricot, and other tree species [20,22]. In the present study, we further characterized the time range of attraction in apple orchards as well as the spatio-temporal effect on lacewing density and biological control of aphids in barley. Both factors are critical in the evaluation of such formulation for a sustainable control of aphids. Together with the fact that the formulation is biodegradable and amenable for mechanical distribution, makes the paste promising for practical application, as reported earlier for similar products [23,30]. The replacement of current aphicides with this novel formulation can support ecological intensification of integrated pest management programs.

The novel biodegradable formulation at the tested dosages triggered significant oviposition and larval presence in comparison to the control. While both parameters were inversely correlated with distance from the formulation, aphid population reduction remained elevated on the neighboring plants, with a possibly longer range effect. The increased presence of *C. carnea* eggs and larvae was most likely causing the significant reduction of both *R. padi* and *S. avenae*. However, we cannot completely rule out a possible repellence of aphid by the volatiles released from the tested formulation. For example, *cis*-jamone and MS were reported to repel aphid alates, including *R. padi* and *S. avenae* [31,32].

In our conditions, a lower number of lacewing eggs and larvae were measured for the higher dollop dose, possibly due to an initial repellent effect. Repellence of lacewings and other natural enemies to high loads of MS has been reported in other studies [33], along with laboratory trials showing oviposition avoidance in the presence of eggs previously laid by conspecific individuals [34]. While this last phenomenon was not observed in the present study, an increase of larval abundance over time was observed. Whether or not the tested formulations are capable of recruiting lacewing larvae from adjacent plants, however, remains unclear.

The novel lacewing formulation released the three loaded volatiles over the tested period. Although the emission from the paste was lower than from the PE bag, and a lower number of lacewings were attracted to it, reduction of aphids was similar. The profile of the additionally released compounds was, however, different. These compounds are most likely breakdown products or impurities derived from synthesis, and are present in lower concentration than the main three components. How these impurities have contributed to differential attraction between the formulations is not clear. Whereas 2-phenylethanol and benzyl alcohol triggers an antennal response together with PAA in both male and female lacewings, benzyl acetate and benzaldehyde responses were not different from water [20]. Lacewing response in the field to 2-phenylethanol has been shown also to be lower than to a blend of MS, PAA, and AA [35]. Adding 2-phenylethanol to the ternary blend did not increase attraction [20]. Benzaldehyde attracted *Chrysoperla plorabunda* (Fitch) with captures, varying with field location [35]. No information is available on the behavioral effect of the other compounds.

Although lacewings were the predominant beneficial insect present in the barley field, ladybirds, hoverflies, and parasitized aphid mummies were also recorded. All ladybird adults observed belonged to the aphidophagous *C. septempunctata*. Therefore, although the larvae have not been identified at the species level, it seems reasonable to estimate that all of them belonged to the same species, and not to fungus feeders or pollen feeders. As lacewings arrived to the field 1–2 weeks earlier than ladybirds and approximately 6 weeks earlier than hoverfly larvae, we suspect that this early arrival was the factor shaping the composition of natural enemies in the experiment. In addition, the presence of lacewing larvae with the artificially added volatiles may have caused aversion to other predators with a reduction of oviposition due to the associated risk of intraguild predation. The semiochemical-mediated specie-specific avoidance of oviposition to conspecific or heterospecific larvae is known for several species of ladybirds and lacewings [34,36]. On this note, the higher amount of aphid mummies in the control could be due to repellence of parasitoids, as in other studies with synthetic volatiles [37], or predation of parasitized aphids by lacewing larvae. In a laboratory study, *C. carnea* and *C. septempunctata* consumed higher numbers of parasitized than un-parasitized aphids [38].

We remark here that early predation, as promoted by the lacewing formulations, is a key factor in suppression of aphid population, avoiding higher densities and potentially associated damages, as already reported for aphids [39].

While applications of HIPVs are promising for pest control, several factors need to be considered before full field transition. For example, attraction of natural enemies to synthetic kairomones in the absence of the associated herbivore prey may lead to their starvation with unpredictable consequences on the ecosystems [11]. This may be especially relevant for lacewing larvae. Further studies should consider directly rewarding recruited adults using (e.g., sugar or protein laced formulations or flower strips within or around the crop). Another issue to consider is the possible attraction of insects belonging to the fourth trophic level, which can reduce the level of herbivore predation in the long term by the targeted beneficials [40]. In order to reduce the negative effect of possible cannibalism among predators due to a low herbivore population level, HIPVs can be combined with non-crop vegetation such as flower strips, providing alternative resources (prey, nectar, and pollen) [41,42]. In our field setting, synthetic volatiles triggered oviposition of lacewings and increased lacewing larvae presence without the introduction of any reward component. Because in our experimental set-up, reduction of aphid infestation was achieved via natural enemy recruitment, we confirmed the possibility of using this method as a step towards a future ecological intensification of cultivated systems. We need to remark that our observations were collected within a landscape with wild inter-field vegetation, where lacewings may have had access to floral resources, overwintering sites, and alternative prey. It thus remains to be tested whether our result can be repeated within a larger and more intense monoculture system. An additional challenge of large-scale deployment of predator attractants is that this approach is drawing on a limited pool of predators across the landscape. While a higher density of predators is likely to occur in the field where attractants are deployed, a dilution effect may take place in neighboring fields where the same predators are pulled from. Accordingly, significant landscape- and population-level questions need to be addressed before this technology can be effectively and ethically utilized on a large scale.

## 5. Conclusions

Although our results show a promising lacewing attraction in both a perennial and an annual cropping system, a measurement of high biological control by lacewing larvae could be observed only in barley. The distribution of aphids in apples is far more uneven and unpredictable than in an annual crop, due to characteristics related to the re-immigration from the secondary host to the overwintering host, as in the case of the rosy apple aphid. Additional experiments are thus recommended to evaluate the potential of the new formulation for biological control of aphids in apples.

The development of semiochemical-based methods, such as the one tested here, can offer innovative and efficient approaches to control pest populations via a selective manipulation of the behavior of beneficial insects. Additional research should support practical integrated pest management guidelines, including threshold value for pests and natural enemies, mechanization of application technique, and the evaluation of local non-crop vegetation as a natural enemy reservoir.

## Figures and Tables

**Figure 1 insects-10-00006-f001:**
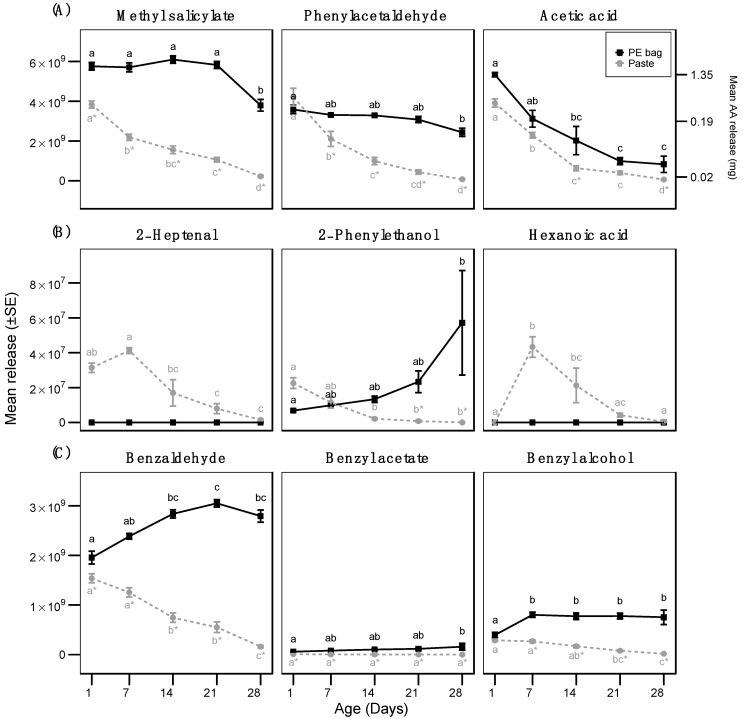
Mean release rate (±SE) of (**A**) methyl salicylate (MS), phenylacetaldehyde (PAA), and acetic acid (AA) from two different emitting devices at five different ages of field exposure (N = 5). The semi-quantitative release of AA is shown on the right y-axis. Letters above points indicate significant differences between ages of the same formulation and * significance between formulations at a given age (LRM and GLS, Tukey’s test, *p* < 0.05). (**B**) Mean release rate (±SE) of 2-heptenal, 2-phenylethanol, and hexanoic acid. (**C**) Mean release rate (±SE) of benzaldehyde, benzyl acetate, and benzyl alcohol. 2-heptenal and hexanoic acid were exclusively found in Paste.

**Figure 2 insects-10-00006-f002:**
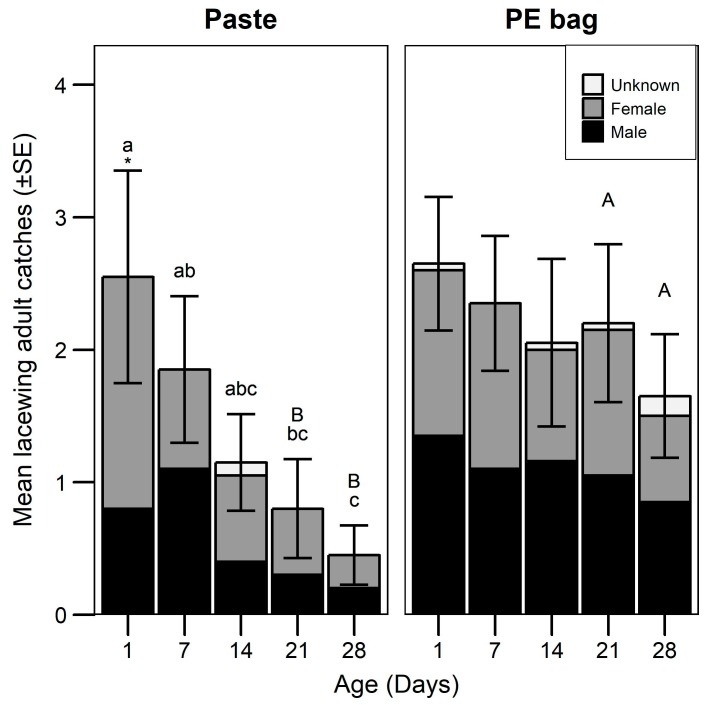
Mean catches (±SE) of lacewing adults in apple orchards by two different emitting devices at 1, 7, 14, 21, and 28 day of field exposure (N = 5). The lower part of the bar indicates the proportion of males, middle the proportion of females, and top the proportion of individuals that could not be sexed. Different uppercase letters above the bar indicates a significant difference of catches between the two formulations at a given age (GLMM, Tukey’s test, *p* < 0.05). Different lowercase letters indicate significant differences between ages within the same formulation (GLMM, Tukey’s test, *p* < 0.05). An * above bars indicate significant higher ratio of females at the corresponding age (GLM, Tukey’s test, *p* < 0.05).

**Figure 3 insects-10-00006-f003:**
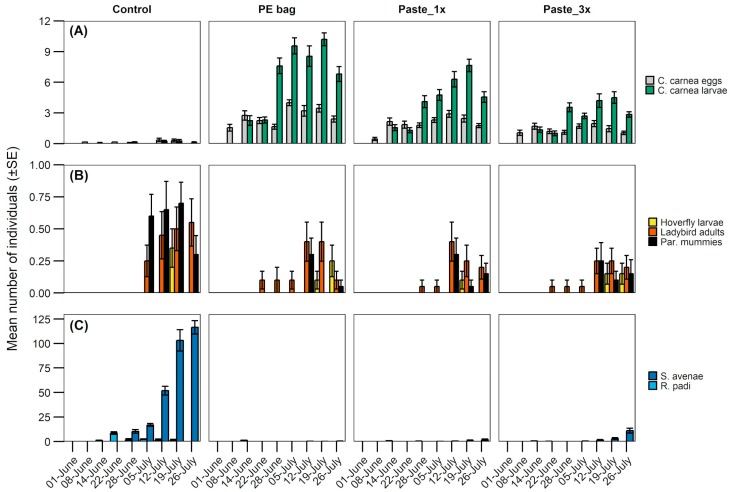
Mean number (±SE) of (**A**) lacewing, (**B**) other natural enemies, and (**C**) aphids observed on three barley plants during the experiment.

**Figure 4 insects-10-00006-f004:**
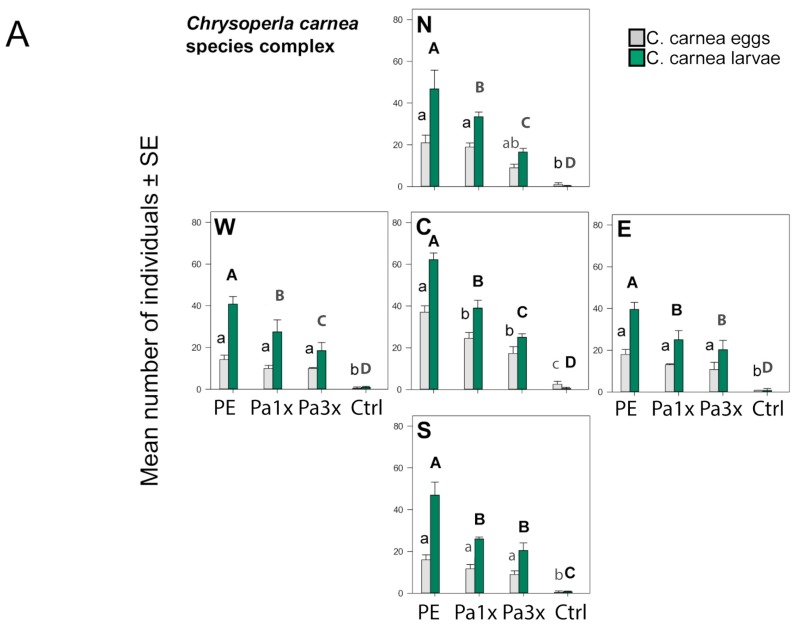
Cumulated number of individuals (±SE) of natural enemies and aphids measured on three barley plants in the central (C) sector of the plot and on plants located at 30 cm north (N), south (S), east (E), and west (W) from the central sector. Different letters above bars indicate significant differences between formulations in each sector for (**A**) lacewing eggs (lowercase) and larvae (uppercase) (GLMM, Tukey’s test, *p* < 0.050), (**B**) ladybird (lowercase), hoverfly larvae (uppercase), parasitoid mummies (bold lowercase) (GLMM, Tukey’s test, *p* < 0.050), (**C**) *S. avenae* (uppercase) and *R. padi* (lowercase) (GLMM, Tukey’s test, *p* < 0.050).

**Table 1 insects-10-00006-t001:** Results of statistical analyses.

Experiment	Model	Statistical Test (Distribution)	Fixed Factor/s	χ^2^./F	df.	*p*-Value
Apple (volatile release)	Methyl salicylate	LRM (Gaussian)	Formulation	967.2	1	<0.001
			Age	59.9	4	<0.001
			Formulation:Age	18.1	4	<0.001
	Phenylacetaldehyde	LRM (Gaussian)	Formulation	120.6	1	<0.001
			Age	40.0	4	<0.001
			Formulation:Age	17.6	4	<0.001
	Acetic acid	GLS (Gaussian)	Formulation	24.8	1	<0.001
			Age	678.9	4	<0.001
			Formulation:Age	2.2	4	0.7
	2-heptenal	LRM (Gaussian)	Age	17.9	4	<0.001
	2-phenyletahnol	LRM (Gaussian)	Formulation	16.8	1	<0.001
			Age	0.6	4	0.641
			Formulation:Age	8.1	4	<0.001
	Benzaldehyde	LRM (Gaussian)	Formulation	820.7	1	<0.001
			Age	17.3	4	<0.001
			Formulation:Age	60.4	4	<0.001
	Benzyl acetate	LRM (Gaussian)	Formulation	226.4	1	<0.001
			Age	0.3	4	0.906
			Formulation:Age	4.3	4	<0.001
	Benzyl alcohol	LRM (Gaussian)	Formulation	316.6	1	<0.001
			Age	7.9	4	<0.001
			Formulation:Age	18.0	4	<0.001
	Hexanoic acid	LRM (Gaussian)	Age	18.2	4	<0.001
Apple (trap catches)	Trap position	GLMM (Negative binomial)	Position	2.4	5	0.795
			Formulation	0.3	1	0.602
			Positoin:Formulation	2.5	5	0.777
	Age vs. Blank (PE bag)	GLMM (Negative binomial)	Age	23.2	5	<0.001
	Age vs. Blank (Paste)	GLMM (Negative binomial)	Age	30.6	5	<0.001
	Lacewing catches	GLMM (Negative binomial)	Formulation	12.2	1	<0.001
			Age	20.0	4	<0.001
			Formulation:Age	8.3	4	0.082
	Sex ratio between formulations	GLMM (Binomial)	Formulation	2.5	1	0.109
			Age	2.5	4	0.645
			Formulation:Age	6.2	4	0.187
	Sex ratio age (Paste)	GLMM (Binomial)	Age	7.1	4	0.133
	Wasp (PE bag vs. Blank)	Fisher’s exact test	Treatment	9.2	-	0.007
	Wasp (Paste vs. Blank)	Fisher’s exact test	Treatment	10.9	-	0.003
	Wasp (PE bag vs. Paste)	Fisher’s exact test	Treatment	1.2	-	0.377
Barley	Lacewing eggs	(GLMM) Poisson	Treatment	137.4	3	<0.001
			Sector	574.4	4	<0.001
			Date	96.5	1	<0.001
			Treatment:Sector	14.3	12	0.279

	Lacewing larvae	GLMM (Negative binomial)	Treatment	408.9	3	<0.001
			Sector	44.1	4	<0.001
			Date	759.7	1	<0.001
			Treatment:Sector	11.4	12	0.499

	Hoverfly larvae	(GLMM) Poisson	Treatment	2.1	3	0.989
			Sector	2.2	4	0.994
			Date	19.0	1	<0.001
			Treatment:Sector	1.6	12	0.999

	Ladybird	(GLMM) Poisson	Treatment	5.6	3	0.133
			Sector	5.6	4	0.429
			Date	67.7	1	<0.001
			Treatment:Sector	12.0	12	0.444

	Parasitoid mummies	(GLMM) Poisson	Treatment	30.1	3	<0.001
			Sector	7.7	4	0.171
			Date	50.4	1	<0.001
			Treatment:Sector	9.1	12	0.697
	*S. avenae*	GLMM (Negative binomial)	Treatment	628.6	3	<0.001
			Sector	3.8	4	0.441
			Date	455.8	1	<0.001
			Treatment:Sector	17.5	12	0.130

	*R. padi*	GLMM (Negative binomial)	Treatment	120.0	3	<0.001
			Sector	2.6	4	0.631
			Date	13.0	1	<0.001
			Treatment:Sector	6.0	12	0.914

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
