# Peer review of "Recruiting on the Spot: A Biodegradable Formulation for Lacewings to Trigger Biological Control of Aphids"

_insects, 2019, doi:10.3390/insects10010006_

Round 1
Reviewer 1 Report
I must inform the authors (as I previously informed the Editors) that I have reviewed a previous version of this manuscript submitted to another journal. I have read the revised version of the paper, and I very much appreciate the efforts made by the authors to take into account the previously given comments. Thanks to them. Indeed, the manuscript is much clearer now. I still have some minor comments that I list below. I think that the paper should be acceptable for publication once these comments have been considered.
L. 85-88: This sentence is strangely built. I suggest making two sentences: the first one stopping after “(Hordeum vulgare)”; and a second one telling the aphid species, and the lacewing species.
Moreover, L.86: I disagree with the fact that “herbivore consumption by the newly hatched larvae” was measured. In reality, abundance of insects (carnivores and herbivores) was measured, and it has been supposed that the reduction of herbivores is due to an increase of carnivores. However predation itself has not been measured. Please correct or delete this part of the sentence.
L. 131: Please delete “s” of “Attraction”.
L. 137-142: The experimental design in apple orchards is still a little difficult to understand. Why not providing a scheme of the experimental design in a Figure? Sometimes, images are more straightforward than long sentences.
L. 146: When I have started reading this part of the Material & Method, I was asking myself: what is the landscape context of this field? And I finally got the information at the very end of the paper (L. 338-340). Please provide information about the landscape context here, and they can still be recalled at the very end as it is the case now.
L. 172: Please provide the reference related to the package lme4. Moreover, reference number into [] must be used for (Length, 2016) and (Fox and Weisberg, 2011).
L. 203-204: “3. Results” must be in bold, and “3.1 Measurement of volatile release” must be in italics.
L. 205: It does not appear in Fig. 1 that “the emission of each volatile decreased”. For example 2-Phenylethanol or Benzaldehyde increased for one of the two device (black line). They decreased for MS, PAA and AA, yes.
Fig. 1: Please indicate on the figure the letters “A, B, C” of its different parts. Moreover, please indicate in a legend what the black line is and what the grey line is. Also, why are there no indications of statistical differences between the two release lines from 2-Heptenal to Benzyl alcohol (the 6 minor compounds)?
L. 223: Did the reference device attract a SIGNIFICANT higher number of lacewings than the new paste? Please specify in the text.
L.224: How can it be affirmed that “the decline of in attraction was comparable”? Is it because the interaction effect between ‘lure’ and ‘age’ was not significant (Table 1)? Please specify in the text.
L.228: Please add a phrase like “In addition to lacewings, …” before “A total of 89 wasps…” to make a transition.
L. 255-259 and L. 265 (Fig. 4 caption): it is not clear if there are ladybird larvae or not. In the paper in general, sometimes only “ladybirds” are mentioned, sometimes “ladybird larvae”. Please specify ladybird adults, or larvae, or both.
L. 310: Please provide the reference where this result appear, after “… did not increase interaction”.
L. 311: Please provide the reference where this result appear, after “… varying with field location”.
L. 317: To me, the abundance of lacewing larvae is very low without the artificially added volatiles (control plots in Fig. 4). I don’t understand why here there is a mention of “with and without” volatiles?
319: Please add reference(s) that help you to hypothesize that the risk of intraguild predation may have repulsed other natural enemies than lacewings to oviposite where lacewings were abundant.
321: Please add a reference stating that lacewing larvae can prey on parasitized aphids.
323: Please use reference numbering into [] for “(Porcel et al 2018)”.
L. 338: Please add an “s” to “observation”.
L. 345: “due to the characteristics related to the cycle of aphids themselves”. This sentence is unclear. Please be more specific: what characteristics? Why is it different in apple orchard compared to barley field?
Author Response
85-88: we think this sentence reads well and it is not complicated at all. Please let us know if you share or not this point. We found similar sentences in many other papers. See for example Knudsen and Tasin 2015 (Basic and applied ecology, introduction).
86: changed
131: we did not find this “s” in “attraction”. Perhaps there is a mismatch with line numbering. We changed at line 133.
137-142: a new figure with the field configuration of the experiment is now added in as S1 figure
146: information on the surrounding landscape is now added in at the end of the paragraph.
172: bibliography added in
203: done
205: true. The sentence is now changed.
Fig 1.: changed
Fig 2: changed
223: Added that the toal trap catches is different
224: Added ref. to fig. 2 with a post hoc test. It is also more explained in the next sentence.
228: done.
255-259: changed.
310: changed.
311 (now 317): added in.
317: removed “without”
319: changed.
321: changed
323: formatted
338: changed.
345: information is now added in to better explain the difference.
Reviewer 2 Report
The manuscript by Pålsson and Thöming shows an interesting study of a biogradable formulation, composed by a mix of synthetic HIPVs, to trigger the biological control of aphids by lacewings in different crops. The study is separated in two sections (crops); first one shows the volatile release of two different devices (Paste and PE bag) and how this formulation induce attraction to lacewings in apple orchards. Second part it shows the attraction induced by both devices to lacewings and the biological control performed by these natural enemies on aphid populations in barley crops.
The first part of the experiment (catches of lacewings in apple orchards) it may need a control (catches without any lure) to identify the direct effect of the lure and devices as occurs with the barley experiment.
It would be interesting to see the data of measurement of volatile release not only for apple orchads. Adding this data also for the barley may allow to the authors to measure the volatile release depending on the temperature.
ABSTRACT
L14: I do not understand this sentence.
INTRODUCTION
L51 I think it is better to say predators instead of carnivores in this context. Please, be consistent throughout all the manuscript.
L52 to 54 Please check the literature of the zoophytophagous insects.
L61 I think it is better to say predators instead of carnivores in this context.
L71 Please be specific. Change several for some examples from the literature.
MATERIAL AND METHODS
L148: 5 meters distance between plots do not seems enough to find difference between treatments. This may affect the differences between the Paste and PE bag.
L186: I could not find the comparison with a “blank” trap in results section.
Fig.1 I think that this figure needs a legend to differentiate the emitting devices. Please write the letters on the figures that you describe in the caption (A.B and C) or remove from the caption. Please, include the x-axis label [i.eg. age of the lure (days)] and the y-axis label (with units). It may help also to include the acronyms on the figure.
Line 228: Please check this information. I do not see any information about wasps catched on apple orchards in Table 1.
Fig. 2 Please specify in the caption what crop are you using for this experiment. It is confusing to me the uppercase letters in this graph. If I understood well, with uppercase letter you are comparing catches between the two formulations at a given age, so you want to compare Paste of 1 day old with PE bag of 1 day old only. If this is correct it seems that age 21 and 28 differ significantly from Paste to PE bag. Why then they have the same letter “B”?. I suggest to do not write a letter when there is no differences (i.eg. Paste 1 day and Pe bag 1 day) and write the letter when there is differences (i.eg. Paste 21 days and PE bag 21 days). I this case I suggest to put the “A” on the higher mean lacewing adult catches and the “B” on the lower one.
Fig. 3 Very interesting figure. It´s shows how important it is to attract the natural enemies before the pest start to grow exponentially (L273).
I suggest to label A, B and C as lacewings, other natural enemies and aphids respectively only in the caption.
L249: Please add (Fig. 4?)
Fig. 4. I think that this figure do not add too much information to the manuscript. The authors use this figure to refer on the text for results and discussion that can be explained with figure 3 only. It could be because the distance between surveys (C N W E S) it is only of 30 cm. However, the first part of the figure (A) shows a bit of the repellent effect for the higher dosage which the authors discussed in the Line 294.
Please explain in the caption what crop are you using in each Figure. I suggest also to include the x-axis label in the “W” and “E” graphs, despite it is repetitive I think it will help the reader.
L291: For the “neighbouring plants”, the survey was done at 30 cm from the lure. Is this enough distance to recall as effective for “neighboring plants”?
L317: How is this?, intraguild predation? Lack of preys/host late in the season?. Please expand this discussion and add some references.
L323: please correct the format of the citation.
Author Response
L14: added “...which induces chemical defences in the plant as well as ...“
L51: changed
L61: changed
L71: added
L148: that is one limitation of our study. It is a compromise between separating treatments but still be within same “population” of lacewings for that block. We wanted to have competition between treatments to assess which is most attractive. The control is further away so it is not influenced by the treatments.
L186: our mistake. added it in text as table 1
Fig1: changed.
L228: wasps are now added in.
Fig2. changed.
Fig3: done
Fig4. we think the figure is showing very well the experimental design and the effect of treatments. We would like to keep it in the paper.
L291: changed.
L317: we discussed this further
L323: DONE
Reviewer 3 Report
The manuscript proposes a new formulation of plant and food volatile compounds embedded in a biodegradable matrix as a mean to recruit aphidophagous natural enemies, mostly lacewings.
Overall, the manuscript is clearly written, with a focused introduction, thorough description of the methods and a clear explanation of the results. However, before recommending it, some parts should be improved. Although one of the aims was to link the aphid suppression with the recruitment of lacewings by the developed lure, this is not fully supported by the results. Below, my comments:
L51: To be consistent within the text, I would use hereafter the term predators rather than carnivores. However, this sentence is somewhat repetitive of L46-48.
L52-54: Please reword the sentence, it is too vague. I would mention about the reliability of HIPVs for generalist and specialist predators and parasitoids.
In L80-81: Unfortunately, this hypothesis has not been properly tested. In fact, the experiments lack a proper control that would have helped to discern whether the low aphid abundance was due to the lacewing presence or to the lure repellence versus the aphids (see also the comment below).
L171-202: I would suggest adding a brief description within brackets of the covariates that have been used in the models. In fact, it is very hard to follow the text, for example, what is the difference between the two covariates “formulation” (L175) and “lure” (L179)? In L93 and L108 is “formulation”, while in L147 is “lure”. Similarly, in L188 and L189. I am guessing from L196-197 that they are synonymous.
In addition, in L177, it is not clear to me why the authors fitted Generalized Least Square models because of non-normality of residuals. To my knowledge, GLSs are used to deal with heterogeneity of the variances. If the Authors have concerns about normality, I would propose applying a different data transformation or, eventually, use a non-parametric test (e.g. Kruskal-Wallis).
L125-130: Please, add one or more citations for this part.
L159: Please specify how non-destructive samplings have been conducted (e.g. with beating trays, sweep netting or visual inspections?).
L205: I would also include the estimated coefficients in table 1, as to provide support for this sentence. However, the sentence is not fully correct because the releases of 2-phenylethanol and benzaldehyde increase over time with PE-bags.
Fig. 1: Please, add the titles for both axes. Also, letters for figs A, B, C are missing. The caption does not tell about line colours – I am guessing from the text that light grey is for the “paste” treatment while black is for “PE bag”. Results from multiple comparisons should be added also for fig. 1B and fig. 1C
Table 1: Please add to the table caption more information about the columns’ description. In addition, info for two covariates that are related to “2-heptenal” and “Hexanoic acid” are missing.
Fig. 3: Please format the picture so than the 12 plots have a similar size.
L254-255: The Authors report the density of ladybird beetles without providing any specific identification. Although aphidophagous ladybirds are supposed to demographically respond to aphid-associated lures (e.g. C. 7-punctata) there are few commonly frequent species that feeds on fungi or pollen (see Hodek and Honek 1996). Therefore, the information about total larval density is practically not very useful. If recorded, I would suggest adding only the data on the aphidophagous species. The density of ladybird adults is also important because of their predatory activity, contrary to C. carnea and hoverfly adults, which are nectar and pollen feeders. If those data have not been collected, I suggest removing from the manuscript the information about ladybirds.
Fig. 4: The figure is very complex, with results of several cross-comparisons. Therefore, it is very hard to follow. I suggest moving to the supporting information and add in a table the estimated parameters from the models, SE and significance of post-hoc comparisons. Also, because there not seems to be lots of differences between the N-S-W-E-C.
L292-293: Please reword here and among the text (in L21; L302-303; L334-336). In fact, evidences of aphid suppression due to the presence of C. carnea eggs or larvae are not properly demonstrated. The lower aphid density might be merely the consequence of the lure repellence against aphid females. For example, induced plant volatiles repelled the two aphids R. maidis and S. avenae in olfactometer bioassays (Pettersson et al., 1994; Bernasconi et al., 1998; Bruce et al., 2003). Unfortunately, the experiments lack a proper control to discern the effect of repellence. For this purpose, it would have been useful to include a treatment with an excluding cage allowing the entrance of aphids but not of lacewings, there are several examples of this methodology in the literature.
Also, the information provided about the aphid density is not very helpful to solve the question. In fact, it is not specified the number of alates vs apterus. In the case that these data have been collected, it would be very helpful to compare the presence of alates in treatment vs. control at the beginning of the season (immigrant adults), as well as during the season (emigrant adults). Indirectly, these data might help to elucidate the effect of lures on the aphid repellence.
L319: There is a huge literature on that, please add one or more citations.
L319-320: But why predation by lacewings and not from ladybirds or syrphids? Please, see for example Meyhöfer and Klug (2002) on the predation activity by the abovementioned predators upon Lysiphlebus fabarum. Probably is more reasonable that the higher parasitoid density in the control might be the consequence of the higher aphid population.
Reference section: Please, abbreviate all the Journal names.
REFERENCES
Bernasconi, M. L., Turlings, T. C., Ambrosetti, L., Bassetti, P., Dorn, S. (1998). Herbivore‐induced emissions of maize volatiles repel the corn leaf aphid, Rhopalosiphum maidis. Entomologia Experimentalis et Applicata, 87: 133-142.
Bruce, T. J., Martin, J. L., Pickett, J. A., Pye, B. J., Smart, L. E., adhams, L. J. (2003). cis‐Jasmone treatment induces resistance in wheat plants against the grain aphid, Sitobion avenae (Fabricius)(Homoptera: Aphididae). Pest Management Science, 59: 1031-1036.
Hodek, I., Honek, A., 1996. Ecology of Coccinellidae. Kluwer Academic Publishers, Netherlands. 464 pp.
Meyhöfer, R., Klug, T. (2002). Intraguild predation on the aphid parasitoid Lysiphlebus fabarum (Marshall)(Hymenoptera: Aphidiidae): mortality risks and behavioral decisions made under the threats of predation. Biological Control, 25(3), 239-248.
Pettersson, J., Pickett, J. A., Pye, B. J., Quiroz, A., Smart, L. E., Wadhams, L. J., Woodcock, C. M. (1994). Winter host component reduces colonization by bird-cherry-oat aphid, Rhopalosiphum padi (L.)(Homoptera, Aphididae), and other aphids in cereal fields. Journal of Chemical Ecology, 20: 2565-2574.
Author Response
L51: carnivore is now replaced by other terms
L52: the attraction of generalist and specialist is now added in
L80: in the field, we observed high number of aphids around the HIPVs dispensers, although lacewings were not detected. Shall we add this information into the text?
L171: lure is now replaced by formulation, except when it means “device”
L177: GLS was used to correct for heterogeneity (added) as you mention but it did not fix the normality and therefore the response was also transformed to square root, which is now in the text.
L125: changed.
L159: visual inspection is now added in
L205: sentence changed. Table S2 is now added.
L254-255: updated with the infromation we had.
Fig. 4: we like to keep fig 4 in the text, since it is the main result of the paper. However, we may move the result of the comparison to supplementary material, if the editor agrees.
L292: No, we didn’t investigate repellent effects on aphids. We stated the possibility in the discussion (L. 294ff). We can add the suggested additional literature here (Pettersson et al., 1994; Bernasconi et al., 1998; Bruce et al., 2003) and state that additional research – as suggested, excluding cages / specification of alates vs apterus to study aphid repellence - is needed here.
We changed the lines and added that aphids could be repelledd + references.
L319: changed.
L320: we discussed this issue in the discussion
Reviewer 4 Report
The manuscript is interesting and well-written, for the most part. The results are useful and provide valuable fodder for subsequent work. I feel that it is acceptable with some edits.
My comments are on the attached manuscript, but there are some questions/concerns I’d like to highlight here:
1. What were the field temperatures during the time that the HIPV emitters were in the field? This can profoundly affect duration of emission, and would be a very useful addition to the paper for future reference.
2. Were the surroundings around the two experimental zones (treatments and control) of the barley trial similar?
3. The three HIPV treatments were quite close together in the small barley plots. How much interference/influence might there have been among the treatments since the lacewings could easily pick and choose?
4. I found no reference to Vespids in Table 1, despite the reference to the Table in the text regarding these predators.
5. The figures are not clear. What are the units of measurement (per plant? Per three plants?)? In Figure 1 the caption fails to indicate which line is which treatment, making it difficult to sort out.
6. In the PE treatment, in particular, it appears that virtually every lacewing egg became a larva, and that there were perhaps more larvae than eggs? How accurate were the egg counts in the barley? Also, given that there were no aphids in the volatile treatments, what were the lacewing larvae eating to persist? In the absence of prey, was the barley crop serving as a sink for the lacewing populations?
7. Although I like the ideas that the authors are pursuing – using volatiles to attract predators – there are definitely some challenges with this approach. The authors note the concern about attracting predators to fields with limited or no prey, which can reduce fitness of the predators and may ultimately hurt predator populations on a landscape scale if the crop system is a sink rather than a source. Timing of attraction seems a critical issue. An additional challenge of large-scale deployment of predator attractants is that this approach is drawing on a limited pool of predators across the landscape. Thus, to draw predators to my fields, I must subtract them from somewhere else. It’s a zero-sum proposition. My neighbors may not like the idea of me taking predators from them. Or perhaps many around me are using the same technology and diluting the pool of predators available to me in the center of the HIPV landscape donut. I think that there are some very significant landscape- and population-level questions that need to be addressed before this technology can be effectively and ethically utilized, and it would be useful for the authors to provide rather more consideration of these issues.

Author Response
1 climate data are now added in
2 information is now added in
3 this was done on purpose, with the aim to evaluate which attractant would perform best in a field situation
4 vespid data are now in the manuscript
5 “per three barley plant” is now added in the legends where relevant
6 Here we hypothesized that larvae from the surroundings might be attracted in addition to the larvae hatched from the eggs at the sampling points. This is mentioned at the end of the second paragraph “Whether or not the tested formulations are capable of recruiting lacewing larvae from adjacent plants remains, however, not clear”
How accurate were the egg counts in the barley? All counted eggs were marked with a small dot on the leaf after counting to ensure an accurate counting. Have been added in the M&M.
Also, given that there were no aphids in the volatile treatments, what were the lacewing larvae eating to persist? ˃ This has not investigated in this study, but we discussed this issue (paragraph 313-341).
In the absence of prey, was the barley crop serving as a sink for the lacewing populations? We are talking about this in the last paragraph of the discussion: “For example, attraction of natural enemies to synthetic kairomones in the absence of the associated herbivore prey may lead to their starvation with unpredictable consequences on the ecosystems [11]”. In conclusion we suggested that natural enemies need to be monitored, as a precaution of depleting the population.
7 very pertinent comment. This is now part of the discussion (last paragraph of discussion).
Round 2
Reviewer 3 Report
The Authors addressed previous comments and now the manuscript is clearly presented. In particular, the method section presents more clearly the covariates that have been used in the model fitting and the model coefficients are now presented in a supplementary table.
Moreover, the discussion section now includes a statement about the possible repellent effect on aphids. Therefore, also the conclusions now make more sense to me.
I have the following minor comments/suggestions:
L161: replace “formulation” with “formulations” and add a dot at the end of the sentence.
L201: parasitic wasps?
L201: change “Fisher exact test” with “Fisher’s exact test”
L223-226: Please, replace the comma with dot as a decimal separator
L240: replace “28 day old” with “28-day-old”
L241: remove “significantly” and move “(Table 1)” after the word “paste”
L242: replace “Fig2” with “Fig. 2”
Table 1, column “Distribution”: actually, “Fisher’s exact test” is a statistical test and not a conditional distribution like Poisson/Binomial, ecc. A possible solution might be relabelling the column header with “Statistical test (Distribution)” and including the model details along the column cells, e.g. “GLMM (Gaussian)”.
L273: Please, add “L.” after Coccinella septempunctata. In addition, it is valuable to add a sentence in the discussion underlying that all the ladybird adults collected belonged to the aphidophagous C. 7-punctata. Therefore, although the larvae have not been identified at the species level, it seems reasonable that all of them belonged to that species, and not to fungus-feeders or pollen-feeders. This would help in interpreting the results from table 1.
L343: remove the comma after “lacewings”
L488-490: Please correct the typos
Author Response
We accepted all the proposed changes.